# Aluminum Bioaccumulation in Reed Canary Grass (*Phalaris arundinacea* L.) from Rivers in Southwestern Poland

**DOI:** 10.3390/ijerph19052930

**Published:** 2022-03-02

**Authors:** Magdalena Senze, Monika Kowalska-Góralska, Katarzyna Czyż

**Affiliations:** 1Department of Limnology and Fishery, Institute of Animal Breeding, Wrocław University of Environmental and Life Sciences, 51-630 Wrocław, Poland; monika.kowalska-goralska@upwr.edu.pl; 2Department of Sheep and Fur Animals Breeding, Institute of Animal Breeding, Wrocław University of Environmental and Life Sciences, 51-631 Wrocław, Poland; katarzyna.czyz@upwr.edu.pl

**Keywords:** aquatic plants, *Phalaris arundinacea* L., aluminum, rivers, water, indicators: MPI, BCF_W_, BCF_B_

## Abstract

This study aimed to determine aluminum levels in reed canary grass *Phalaris arundinacea* L.) in rivers in southwestern Poland—Bystrzyca, Strzegomka, and Nysa Szalona, together with their tributaries. The samples were collected in spring and autumn 2015–2018. The highest amounts of aluminum were recorded in the Nysa Szalona, and the lowest in the Bystrzyca. During the four-year cycle of studies, the highest values were recorded in the last year, and the lowest in the first year. The highest amounts of aluminum were found in all three rivers in the lowland tributaries. In the main rivers, higher amounts of aluminum were found at the mouth of the Nysa Szalona and Strzegomka reservoirs, while the opposite situation was found for the Bystrzyca. Higher aluminum contents were recorded in autumn than in spring, and the values of BCF_W_ (aluminum bioaccumulation factor in relation to water) and BCF_B_ (aluminum bioaccumulation factor in relation to bottom sediments) coefficients were also higher. The MPI (metal pollution index) was arranged in a series: Bystrzyca < Strzegomka < Nysa Szalona, while the degree of pollution was high for Bystrzyca and very high for the other two rivers. The variability in Al levels may be attributed to pollution level in the catchments, but also to successive modernization works carried out in the beds of the main rivers and their tributaries. All these works were carried out in a variable way and often covered only a fragment of the riverbed; therefore, the consequences of activity may have been visible in the catchment but not necessarily in the same vegetation cycles.

## 1. Introduction

Aluminum is the main building block of the Earth’s crust, and its level in the lithosphere amounts to 7.91%. This element occurs on the third oxidation degree, its ionic radius is small, and therefore it is characterized by a strong electric charge. Aluminum does not change its oxidation state Al^+3^; however, depending on the reaction, it may be present in dissolved or undissolved form, as a colloid, ion or in organic-mineral combinations. The presence of aluminum is also found in atmospheric air, and its amount depends on natural dusting, which is enriched by anthropogenic pollution, while its concentration in flowing waters reaches, on average, 0.064 mg·dm^−3^ [1,2,3,4,5,6].

Aluminum is commonly found in plants, and its content may range from 0.2 to even 1000 mg·kg^−1^ [7]; however, this level varies with species, soil type, and environmental conditions. Aluminum availability to plants is dependent on soil pH, which means that an increase in soil acidity is accompanied by an increase in the number of monomeric forms of aluminum, which are the most bioavailable to plants, but at the same time unfortunately toxic [4]. It has been observed, however, that small amounts of aluminum have a beneficial effect, activating the action of certain enzymes and regulating the physical state of plasma colloids, while excessive amounts of aluminum have deleterious effects on plants. The complex mechanism of aluminum toxicity involves impeding the uptake of major nutrients (P, Ca, Mg, K, N) [1,2]. Symptoms of toxic effects can be mainly observed in the underground parts of the plant; they are difficult to identify in aboveground parts due to the fact that aluminum ions are transported outside the root only in small amounts [2,4].

Aquatic vegetation, whether rooted in sediment or not, acts as a filter for contaminants present in the aquatic environment. Submerged and unrooted plants absorb these compounds with their entire surface, while rooted emergent plants—in addition to their contact with water—also absorb nutrients from bottom sediment and from atmospheric deposition. The richness of the flora world is diversified, and as a rule the flora in mountain river sections is much poorer than in lowland sections, which is related to the scarcity of potential areas that could become substrates for plants, the small amount of sediment, and the relatively fast flow of water. More vascular submerged and emergent plants may be found in the foothill and lowland sections of rivers. Apart from environmental conditions, a main obstacle to macrophyte establishment in the riverbed is the adaptation of rivers to human needs, manifested in the form of morphological transformations. These changes are made, most often, in foothill regions and in areas densely populated by people, i.e., in cities, and their main function is to protect against floods. The works carried out in the riverbed relate to the creation of transverse and longitudinal development, which constitutes the regulation of the river, modification of the banks and bottom (removal of sediments, concreting), and depriving the river of its natural character. The riverbeds are excessively straightened, which makes the water flow quicker and limits the presence of unrooted aquatic vegetation. The effect of these treatments is a small variation in the depth and width of the river. Under such conditions, rooted aquatic vegetation has difficulty inhabiting the river, resulting in a reduction in its number and species abundance. As a further consequence, the reduced number of plants are less able to filter the pollutants present in rivers [8,9,10].

In this study, rivers subjected to such modifications that flow to dam reservoirs, from which water is collected for water supply systems of urban agglomerations, were included in the research. The aim of the study was to determine the accumulation of aluminum in aquatic plants extracted from these rivers over a cycle of several years of seasonal changes, indicating the possibility of aluminum retention in the river ecosystem, and thus showing the condition of the natural environment in this area.

## 2. Materials and Methods

### 2.1. Study Area

The study area was located in southwestern Poland (Lower Silesia Province): N50°38′10.1652″–N51°4′31.7745″ and E16°3′54.4715″–E16°25′1.4097″ and included three rivers together with their tributaries: Strzegomka (5 sites), Bystrzyca (10 sites) and Nysa Szalona (14 sites) (Figure 1).

The following sampling sites were established on the Strzegomka River: (1) Strzegomka below the springs in Nowe Bogaczowice (upland silicate stream with coarse-grained substrate—western, type 4, surface water body status—unitary surface water body—artificial status) (N50°50′14.5978″; E16°7′49.845″); (2) Polska Woda (N50°52′48.0601″; E16°11′56.4194″); (3) Sikorka (N50°51′47.2613″; E16°13′21.3918″); (4) Czyżynka (N50°52′15.8303″; E16°14′29.8332″); (5) Strzegomka—outlet to Dobromierz Reservoir—upland silicate stream with coarse-grained substrate—western, type 4, surface water body—artificial (N50°53′11.1994″; E16°13′58.4707″) [11].

Sites within the Bystrzyca River: (1) Bystrzyca below the springs in Wrześnik—upland silicate stream with coarse-grained substrate—western, type 4, surface water body status—artificial (N50°38′10.1652″; E16°24′5.7915″); (2) Złoty Potok (N50°38′29.3697″; E16°24′41.0163″); (3) Kłobia (N50°40′9.374″; E16°23′27.0131″); (4) Otłuczyna (N50°40′36.2015″; E16°22′46.8444″); (5) Potok Marcowy Duży (N50°41′5.2762″; E16°22′32.3218″); (6) Złota Woda (N50°41′4.2973″; E16°22′11.0015″); (7) Rybna (N50°41′49.8085″; E16°21′58.1784″); (8) Jaworzynik (N50°43′25.8799″; E16°23′56.5218″); (9) Walimianka (N50°43′49.9381″; E16°24′15.0612″); (10) Bystrzyca River at the mouth of the Lubachów Reservoir—an upland silicate stream with coarse-grained substrate—western, type 4, surface water body status—artificial (N50°45′5.8065″; E16°25′1.4097″) [11].

On the Nysa Szalona, the following sites were identified: (1) Nysa Szalona below the springs in Domanów—upland silicate stream with coarse-grained substrate—western, type 4, surface water body status—natural (N50°51′38.8261″; E16°3′54.4715″); (2) Kocik (N50°52′15.4891″; E16°4′5.9042″); (3) Ochodnik (N50°53′37.1718″; E16°5′59.7672″); (4) Sadówka (N50°55′58.609″; E16°10′11.3627″); (5) Czyściel (N50°57′49.4252″; E16°13′57.6982″); (6) Radynia (N50°58′56.648″; E16°14′13.9202″); (7) Nysa Mała—upland carbonate stream with coarse-grained substrate, type 7, surface water body status—natural (N51°0′10.455″; E16°12′26.0825″); (8) Puszówka (N51°2′30.3945″; E16°11′39.425″); (9) Jawornik (N51°2′57.6884″; E16°10′52.4584″); (10) Księginka (N51°3′17.4033″; E16°10′11.2082″); (11) Starucha—upland silicate stream with fine-grained substrate—western, type 5, surface water body status—natural (N51°4′31.7745″; E16°9′17.7528″); (12) Rowiec (N51°4′22.844″; E16°8′27.5419″); (13) Męcinka (N51°4′29.2507″; E16°7′28.5247″); (14) Nysa Szalona outlet to the Słup Reservoir—a small silicate upland river—western, type 8, surface water body status—artificial (N51°4′29.2507″; E16°7′28.5247″) [11].

The Strzegomka River is a second-order river, a left-bank tributary of the Bystrzyca River (Table 1). The river catchment consists of grasslands, agricultural areas, and small rural buildings. The catchment area is dominated by podzolic, brown podzolic, alluvial soils, and acidic soils. A dam reservoir was built in 1988 on the river (62.00 km) in Dobromierz, and it performs the function of retention (reduction in flood waves) and municipal water supply for the Świebodzice region [11,12].

The Bystrzyca River is a second-order river, a left-bank tributary of the Odra River (Table 1). The river basin is made up of agricultural, forest and grassland areas. The dominant soils are podzolic and brown soils and deluvial deposits. A sewage treatment plant and a waste dump are located in the catchment area, as well as two larger towns—Głuszyca and Jugowice. A dam reservoir was built in 1918 in Lubachów, 78.00 km along the river, which performs retention, energetic, water-supply, and municipal functions for the Dzierżoniów region [11].

The Nysa Szalona is a third-order river, a right-bank tributary of the Kaczawa river (Table 1). The river basin consists of agricultural areas, forests, and grasslands. The soils present in the catchment are podzols, brown soils, and alluvial soils. In the catchment area, there are two sewage treatment plants located in Wolbromek and Jawor, as well as an aggregate mine and two larger towns of Bolków and Jawor. A dam reservoir was built in 1984 in the village of Słup, on 8.20 km of the river, playing the role of a retention reservoir and a municipal water supply for the region of Legnica [8,11,13,14].

The land use along all three rivers is quite similar: rural areas, wastelands, forests, small towns with currently limited economic management, poorly regulated water and sewage management limited to larger towns. A significant difference between the rivers is the landform (lowland, upland, and mountain catchment), which can affect the leaching of aluminum from soils.

### 2.2. Material

Monocotyledonous emergent plants (helophytes), reed canary grass (*Phalaris arundinacea* L.), growing on river banks were collected for the study [15]. The plant used for the study is a fast growing, perennial plant characterized by early-season growth, high physiological tolerance, and a large range of applications. It has excellent properties from a combustion point of view, and thus is valued due to its possible application as an energy crop. It can be also used in the production of biogas, ethanol, paper or pulp, as well as for the manufacturing of chemical raw materials [16,17,18].

Plants were collected from the Nysa Szalona, Strzegomka and Bystrzyca rivers and their tributaries (Figure 1) [19,20]. From the main rivers, plants were collected below the sources and at their mouths in dam reservoirs (Słup, Dobromierz, Lubachów), and from the tributaries 50 m before their mouth to the main rivers. The research cycle covered the years 2015–2018, and the samples were randomly collected twice a year at the beginning of the growing season (May) and by its end (October).

Plants were collected whole considering root, stem, leaves, and inflorescence. Immediately after collection, they were washed with river water at the collection site. The plants were then dried at room temperature of 22 °C to an air-dry condition. After drying, macrophytes were cut, crushed, and homogenized [21].

### 2.3. Aluminum Content Determination

For the determination of aluminum content, 0.5 g of air-dry and homogenized sample was weighed in an HP-500 Teflon dish (CEM Corporation, Matthews, NC, USA). After adding 10 cm^3^ of concentrated HNO_3_ (Sigma-Aldrich, Poznań, Poland), the samples were left at room temperature for 24 h. They were then placed in a Mars 5 microwave oven (CEM Corporation, Matthews, NC, USA) and mineralized using a 3-stage mineralization. After cooling to room temperature, the mineralizates were transferred to test tubes and diluted to 25 cm^3^ with distilled water. Total aluminum in plants was determined by electrothermal atomic absorption spectrometry (ETAAS) using a Spectra AA-110/220 from Varian (Australia) [22,23].

In total, 232 plant samples were collected. Results are given in mg∙kg^−1^ on a dry weight basis. Test results were verified with certified reference materials—IAEA-336- International Atomic Energy Agency—Analytical Quality Control Services Austria and CRM 482—Commission of the European Communities, Community Bureau of Reference—BCR.

Results for water and bottom sediments published in earlier works were used to calculate the values of the following coefficients [24,25].

The potential for aluminum accumulation in aquatic plants was determined by:

Aluminum bioaccumulation factor BCF_W_ as a ratio of its content in aquatic plant C_P_ to its concentration in water C_W_ [26]:$${\mathrm{BCF}}_{W}=\frac{C_{P}}{C_{W}}$$
aluminum bioaccumulation factor (BCF_B_) as a ratio of its content in aquatic plant (C_P_) to its concentration in bottom sediment (C_B_) [26]:$${\mathrm{BCF}}_{B}=\frac{P}{C_{B}}$$

The assessment of the state of plants with aluminum was carried out using the metal pollution index (MPI) [27]:MPI = (Cf_1_ × Cf_2_…Cf_n_)^1/n^

where Cf_1_, Cf_2_…Cf_n_—concentration of metals.

MPI values less than 2 indicate no impact on pollution degree, values 2–5 indicate very low impact, 5–10 low impact, 10–20 moderate impact, 20–50 high impact, 50–100 very high impact, and above 100 the highest impact.

### 2.4. Statistical Analysis of the Results

Analysis of the results was performed using Microsoft Office Excel 2019 and Statistica 13.0. Calculations were performed using R version 3.6.0. The Shapiro–Wilk test was performed to verify the normality of the distribution.

Spearman correlations were used due to the distribution of samples. Spearman correlation was calculated in Statistica program, and box and whiskers plots were also created in this program. All statistically significant differences were calculated at *p* < 0.05. Due to the data being defined as having a non-normal distribution, the Kruskal–Wallis test with post-hoc analysis was used. An attempt was made to determine the value allowing the data to be divided into two groups differing in a statistically significant manner. The results are presented when such a value could be determined.

The PCA test using r-statistics was applied in order to visualize the differences between the groups (RStudio Version 1.1.442—© 2022–2018, RStudio, Inc., Boston, MA, USA). It was performed on the basis of all data and presented regarding the differences in the parameters of the examined rivers depending on the year, season of research, and river.

## 3. Results and Discussion

### 3.1. Aluminum in Aquatic Plants

Generally, the aluminum levels in reed canary grass showed the lowest aluminum content of 1.25 mgAl·kg^−1^ in reed canary grass sampled from the Kłobia River (a tributary of the Bystrzyca River) in spring 2018, and the highest of 3044.54 mgAl·kg^−1^ in the Sikorka River, tributary of the Strzegomka River in autumn 2018 (Table 2 and Table 3). In all three studied rivers (Nysa Szalona, Bystrzyca, Strzegomka) over the four years (2015–2018), the highest values were recorded in the last year of the study, and the lowest in 2015 and 2017 (Table 2 and Table 3; Figure 2 and Figure 3).

Generally, no differences were observed between the rivers, but the differences were evident between the seasons of the year. Taking into account the Al content in plants, a significant difference in all studied rivers was observed in 2017 compared to other years, and these differences were statistically significant. This situation can be explained by the difference in hydrological and meteorological conditions, as 2017 was classified as a warm year, but at the same time wetter than the other study years [28]. On the other hand, in 2015 low values were found in the Bystrzyca and Strzegomka rivers, while in the Nysa Szalona they were higher at the same time. This cannot be explained by the prevailing meteorological conditions from the Lower Silesia area, and can only be explained by long-term control studies.

Among all the samples, the highest amounts of aluminum were found in plants originating from the Nysa Szalona River and its tributaries (mean 320.24 mgAl·kg^−1^), lower amounts were found from the Strzegomka River (mean 279.25 mgAl·kg^−1^), and the lowest from the Bystrzyca River (mean 212.44 mgAl·kg^−1^) (Figure 4). A similar trend was observed in the case of plants originating only from the tributaries.

Among the tributaries of the Nysa Szalona River, in spring, the highest amounts of aluminum were recorded in plants growing on lowland and downstream tributaries, and in autumn on upland and midstream ones. The entire range of values was from 2.11 mgAl·kg^−1^ (Męcinka, spring 2017) to 2812.89 mgAl·kg^−1^ (Ochodnik, autumn 2018) (Table 3). Higher values were recorded in the Bystrzyca tributaries draining into the lower section than in the upper section. The entire range of values was from 1.25 mgAl·kg^−1^ (Kłobia) to 2022.55 mgAl·kg^−1^ (Jaworzynik) (Table 3). Among only three tributaries of the Strzegomka River, the lowest aluminum values were found in its first tributary, Polska Woda. The general picture showed an increase in plant aluminum content in successive tributaries with the direction of water flow. The whole range of values was from 3.55 mgAl·kg^−1^ in Polska Woda in spring 2018 to 3044.54 mgAl·kg^−1^ in Sikorka in autumn 2018 (Table 3).

Aluminum levels in vegetation growing on the Nysa Szalona, Bystrzyca and Strzegomka ranged from 2.36 mgAl·kg^−1^ (Bystrzyca springs, spring 2018) to 875.75 mgAl·kg^−1^ (Nysa Szalona, mouth to the reservoir, autumn 2018). It was found that in the Nysa Szalona and Strzegomka, the average aluminum level at the mouth to the reservoir was higher than at the site below the springs. The opposite was the case for the Bystrzyca River, where a higher amount of aluminum was recorded below the springs. In conclusion, the content of aluminum in plants was correlated with the river (r = 0.07) and with the sampling site (r = 0.08), but the correlation was weak (Spearman correlation).

The variability of seasons was also reflected in the levels of aluminum present in aquatic plants. In all studied rivers, aluminum content was significantly higher in autumn than in spring (Figure 5). This was also confirmed by a very high Spearman correlation (r = 0.75).

### 3.2. Metal Pollution Index (MPI) of Aquatic Plants with Aluminum

In order to compare the content of aluminum in samples from different sites, an index of aquatic plant contamination with aluminum (MPI) was used. The highest and lowest values were recorded for the Nysa Szalona River (MPI = 103.35 and MPI = 10.21, respectively) (Table 4). In the four-year cycle of the study, higher MPI values were recorded in 2016 and 2018 than in 2015 and 2017. The overall picture of MPI in aquatic plants was arranged in the series Bystrzyca < Strzegomka < Nysa Szalona, and was the same for water and sediment [24,25], i.e., Strzegomka < Bystrzyca < Nysa Szalona (Table 5).

The index of plants contaminated with aluminum in all three study rivers was lower at the site below the spring and higher at the mouth to the reservoir (Table 6). In all rivers, the values at the site below the springs were very similar (MPI = 37.35−39.34). At the mouth to the reservoir, the highest, two-fold increase was recorded for the Nysa Szalona.

### 3.3. Bioaccumulation of Aluminum in Aquatic Plants in Relation to Water (BCF_W_)

Aluminum accumulation in aquatic plants reached the highest values in reed canary grass growing in the bed of the Nysa Szalona River and its tributaries (Table 2 and Table 3). Lower and similar amounts were accumulated by plants from the Bystrzyca River and the lowest from the Strzegomka River and its tributaries. For all obtained results, the range of BCF_W_ values was from 10.83 to 17473.74 (Table 2 and Table 3).

In the Nysa Szalona and Bystrzyca Rivers, the accumulation coefficient was higher at the reservoir mouth than at the spring, and in the Strzegomka River it was the opposite. Over four years of study, at the site below the springs in the Nysa Szalona and Strzegomka the lowest values were recorded in 2017 (this year differed significantly from the others) and the highest in 2018 (Table 2). In the Bystrzyca River, the lowest values occurred in 2018 and the highest in 2016. At the mouth to the reservoirs, the lowest BCF_W_ values were noted in 2017 for all three main rivers and the highest in 2018 for the Nysa Szalona and Bystrzyca Rivers, and in 2016 for the Strzegomka.

Aluminum bioaccumulation index was higher at all study sites in the autumn compared to the spring period, indicating aluminum uptake from the aquatic environment throughout the growing cycle (Table 3).

Among the tributaries of the Nysa Szalona River, the highest accumulation occurred in Męcinka (a tributary of the lower reaches of the river) in spring, and in Ochodnik (a tributary of the upper reaches) in autumn (Table 3). Within the Bystrzyca, the highest accumulation in spring occurred in Kłobia (upstream tributary), and in autumn in the downstream part of the river in Jaworzynik. In the Strzegomka River, in spring the highest BCF_W_ value was found in Czyściel (lower reaches), and in autumn in Sikorka (middle reaches).

### 3.4. Bioaccumulation of Aluminum in Aquatic Plants in Relation to Bottom Sediments (BCF_B_)

Aluminum accumulation in aquatic plants in relation to bottom sediments was the highest within the Nysa Szalona River and its tributaries. Much lower values were found in the Bystrzyca River, and the lowest in the Strzegomka River. The whole BCF_B_ range was from 0.0001 to 71.98 (Table 2 and Table 3).

In the Nysa Szalona, a higher accumulation was recorded at the reservoir mouth, and a lower one at the spring. The opposite situation was found for the Strzegomka, and in the Bystrzyca the values were very close to each other. During the four-year study cycle in the Bystrzyca River, the site below the springs and at the reservoir mouth had the lowest values in 2017 and the highest values in 2018 (Table 2). In the Strzegomka River in 2018, and in the Nysa Szalona River in 2015, the lowest values were recorded both at the springs and at the mouth to the reservoir. For the maximum values, no such regularity was found.

At all studied sites, the values were higher in autumn than in spring (Table 3), and the differences were statistically significant. Among the tributaries of the Nysa Szalona River in spring, the lowest values were recorded in the upper reaches of the river, while this was much higher in the middle reaches, and decreased again in the lower reaches (Table 3). In autumn, high values were recorded for the first four tributaries in the upper reaches of the river, while for the others the values were lower and quite varied. Irrespective of the season, the next tributaries of the Strzegomka River were characterized by higher values of the aluminum accumulation index (Table 3). In the case of the Bystrzyca River, a slight regularity of the increase in values with the course of the river in both seasons was observed (Table 3).

Aluminum levels found in reed canary grass in this study, when compared with the literature data, were in the range of average values both within Poland, other European countries, and outside of Europe. In Poland, very similar values were found in rivers and small bodies of standing water in the western part of the country in areas subjected to agricultural and industrial activities (range 17–3124 mgAl·kg^−1^) [9,29].

Data on higher and lower ranges of plant aluminum concentrations are much more frequent in the literature. Lower values (400–1000 mgAl·kg^−1^) were found in macrophytes studied in the same rivers (Strzegomka and Nysa Szalona) [30]. Lake hydrophytes from this region of Poland, influenced by industrial areas, were also characterized by a similar range of values (max. 354 mgAl·kg^−1^, BCF = 7420) [31]. Even lower contents were found in aquatic plants of Karkonosze (98–248 mgAl·kg^−1^) [32]. Such low contents were recorded not only in the south of Poland, but also in rivers in Pomerania (0.583–1077 mgAl·kg^−1^); moreover, the values of the accumulation factors were higher there (max. BCF = 70132) [33]. The same was true in plants taken from Pomeranian lakes, where the range was 0.04–33.24 mgAl·kg^−1^, and BCF = 16619 [34].

On the other hand, higher values were recorded in the industrial area (Krakow agglomeration) influenced by mine emissions, where aluminum levels reached a maximum of 8000 mgAl·kg^−1^ [9]. Higher values were also recorded in the Dobra River near Wroclaw (7178 mgAl·kg^−1^) with a bioaccumulation factor reaching the maximum BCF = 91099 and higher values recorded in spring than in summer [35]. Pollutants of a similar nature have been recorded in German mountainous regions, where atmospheric deposition from the industrialized region reaches water bodies there (594–26,200 mgAl·kg^−1^) [30]. High values have also been recorded outside Europe, but they are mainly related to areas under strong anthropopression and the associated acidification of streams and lakes (101–64,422 mgAl·kg^−1^) [36,37,38,39]. In the literature, there are descriptions of reservoirs characterized by low levels of aluminum in hydromacrophytes despite occurring in areas strongly influenced by anthropopression. This was the case, among others, in Turkey (max. 167 mgAl·kg^−1^), in Sicily in Italy (13–3153 mgAl·kg^−1^, BCF = 0.01–0.28), and in Brazil (max. 600 mgAl·kg^−1^) [10,40,41,42,43].

## 4. Conclusions

Aluminum levels in the reed canary grass studied were within the limits of average values for areas moderately affected by pollution. The highest amounts of aluminum were recorded in the Nysa Szalona River, and the lowest in the Bystrzyca River. During the four-year cycle of the study, the highest values were recorded in the last year, and the lowest in the first year. The highest amounts of aluminum were found in all three rivers in the lowland tributaries. In the main rivers, higher amounts of aluminum were found at the mouth of the Nysa Szalona and Strzegomka to the reservoirs, while this was the opposite for the Bystrzyca.

Higher aluminum contents were recorded in the autumn than in the spring, and the BCF_W_ and BCF_B_ indices were also higher. The MPI index was arranged in a series: Bystrzyca < Strzegomka < Nysa Szalona, while the degree of pollution for Bystrzyca was high, and for the other two rivers very high.

The highest aluminum level found in the Nysa Szalona may be related to the largest catchment area above the reservoir, although this rule does not apply to the Bystrzyca, which has a larger catchment area than the Strzegomka, and aluminum values were lower there. Such changes should rather be attributed to the level of pollution in the catchments of these rivers, but also to successive modernization works carried out in the main riverbeds and their tributaries. All these works were carried out in a variable way and often covered only a fragment of the riverbed; therefore, the consequences of activity may be visible in the catchment but not necessarily in the same vegetation cycles.

Significant dissimilarity was observed in 2017 for all rivers. This year, compared to the others, was unusual in terms of climatic conditions: the warmest, but at the same time the wettest. This is confirmed by the data for precipitation amounts from the spring months (higher precipitation) and autumn months (lower precipitation) for the whole area of Lower Silesia [28].

Aluminum accumulation in reed canary grass involves the retention of aluminum in plant tissues. This is particularly important in areas where water is collected for drinking. Aluminum ions that are present in water, when initially used a primary product and later used as a raw material in water production plants, should not reach high concentrations. Therefore, it is important that aquatic vegetation present in water bodies—here, rivers—both submerged and emerged, such as the reed canary grass discussed here, can act as a bioaccumulator and absorb aluminum compounds. The amounts that are thus accumulated in nature will not need to be removed by humans during the water cycle. In addition, the accumulation of aluminum in rivers from the source to the mouth reduces the flow of aluminum in the catchment. In this case, it inhibits the flow of aluminum in the Oder River basin and, further, in the Baltic Sea basin. Hence, the role of hydromacrophytes is extremely important in the absorption of pollutants, including aluminum.

## Figures and Tables

**Figure 1 ijerph-19-02930-f001:**
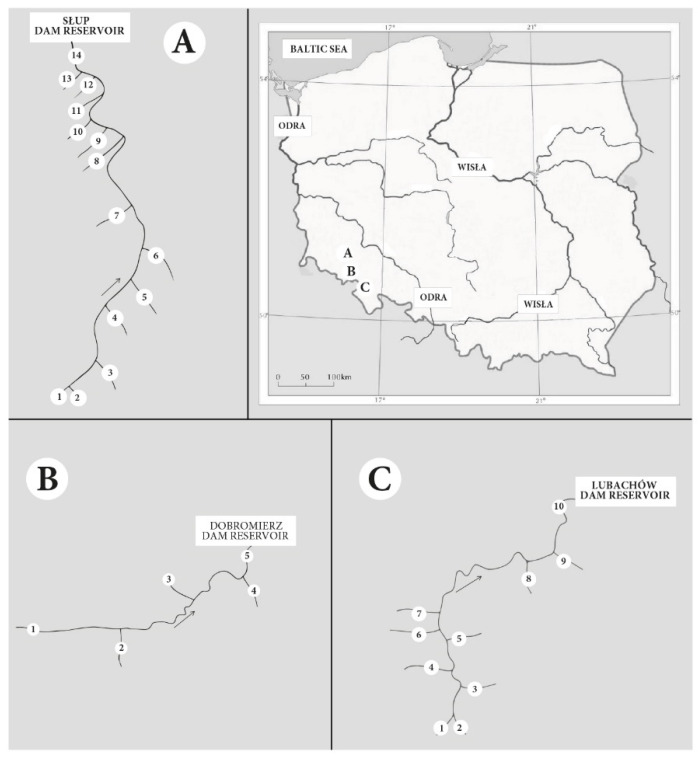
Location of the study area: (**A**) Słup reservoir—research sites on the Nysa Szalona River and its tributaries; (**B**) Dobromierz reservoir—research sites on the Strzegomka River and its tributaries; (**C**) Lubachów reservoir—research sites on the Bystrzyca River and its tributaries.

**Figure 2 ijerph-19-02930-f002:**
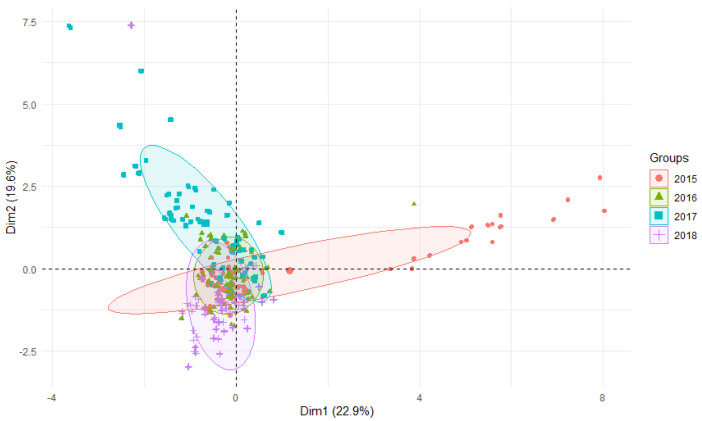
PCA plot 2D showing clustering of Al concentration in plants from rivers across 29 sites and 4 years (2015–2018).

**Figure 3 ijerph-19-02930-f003:**
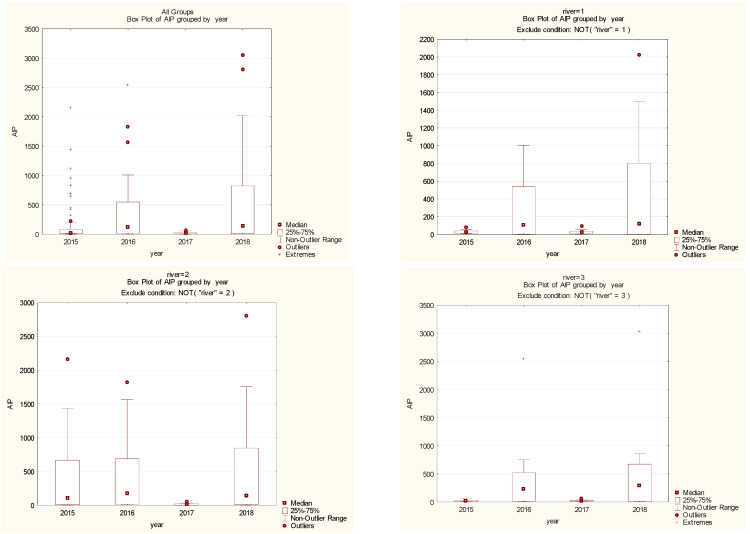
Al concentration in plants from rivers across 29 sites depending on year (1—Bystrzyca, 2—Nysa Szalona, 3—Strzegomka).

**Figure 4 ijerph-19-02930-f004:**
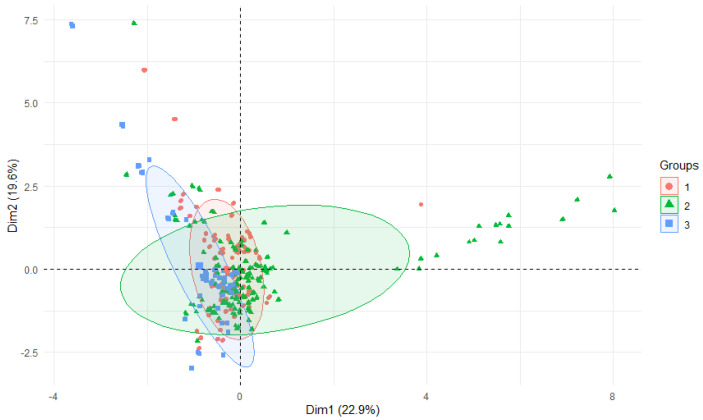
PCA plot 2D showing clustering of Al. concentration in plants from rivers across 29 sites and 3 features (1—Bystrzyca, 2—Nysa Szalona, 3—Strzegomka).

**Figure 5 ijerph-19-02930-f005:**
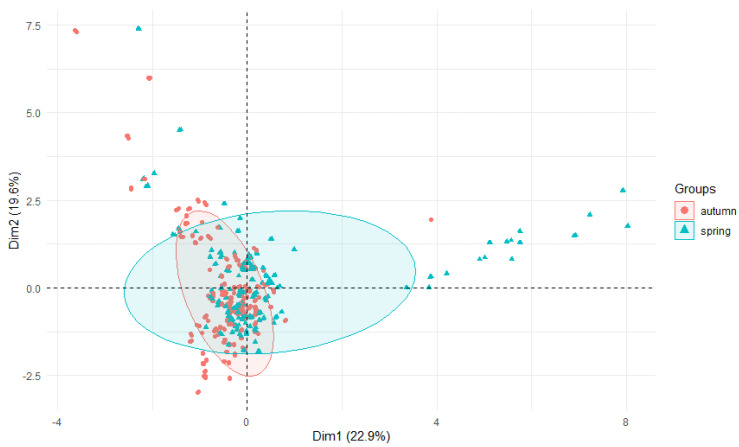
PCA plot 2D showing clustering of Al concentration in plants from rivers across 29 sites and 2 seasons.

**Table 1 ijerph-19-02930-t001:** Characteristics of the Strzegomka, Bystrzyca and Nysa Szalona rivers [8,11,12,13,14].

Characteristics of the Rivers	River		
	Strzegomka	Bystrzyca	Nysa Szalona
Length (km)	74.70	95.20	51.00
Catchment area (km^2^)	555.00	1767.80	443.10
Springs, altitude m above sea level	Trójgarb 692.00	The Suche and Sowie Mountains 618.00	Mount Pustelnik 628.00
Reservoir location m.a.s.l./reservoir type	300–423 lowland and upland	400–500 upland	165–257 lowland
Catchment area above the dam reservoir (km^2^)	70.32	130.69	374.81
Tributaries above the reservoir: left-bank	Sikorka	Otłuczyna, Złota Woda, Rybna	Męcinka, Rowiec, Starucha, Jawornik, Puszówka, Nysa Mała, Kamiennik
Tributaries above the reservoir: right-bank	Polska Woda, Czyżynka	Złoty Potok, Kłobia, Potok Marcowy Duży, Jaworzynik, Walimianka	Ochodnik, Sadówka, Czyściel, Parowa, Kocik

**Table 2 ijerph-19-02930-t002:** Aluminum content (mgAl·kg^−1^ on dry weight basis) in aquatic plants over a four-year study cycle.

Site/Material			2015	2016	2017	2018
			Min–Max $\bar{x}$ ± SD			
Nysa Szalona	Below the springs	P	13.24–423.46 218.35 ± 205.10	10.25–325.10 172.39 ± 162.38	3.15–24.75 13.97 ± 10.72	6.41–562.78 284.40 ± 277.98
			3.15–562.78 172.28 ± 215.39			
		BCF_W_	78.64–3846.13 1956.64 ± 1876.74	103.81–2131.36 1064.65 ± 960.28	14.63–142.02 78.34 ± 63.12	106.34–6045.50 3070.36 ± 2963.77
			14.63–6045.50 1542.50 ± 2127.80			
		BCF_B_	0.0056–0.2028 0.1042 ± 0.0985	0.0002–25.96 12.36 ± 12.38	0.0413–0.2224 0.1255 ± 0.0843	0.0006–0.2923 0.1463 ± 0.1457
			0.0002–25.96 3.1848 ± 8.1486			
	Tributaries	P	5.73–2159.75 400.69 ± 549.99	5.96–1825.44 414.58 ± 513.10	2.11–62.78 14.19 ± 13.26	4.74–2812.89 509.91 ± 688.27
			2.11–2812.89 334.84 ± 544.01			
		BCF_W_	22.78–17,473.74 2982.60 ± 4478.19	68.24–13,333.54 2748.24 ± 3565.82	10.83–358.97 61.8798 ± 71.51	62.74–30,565.43 5679.84 ± 7497.01
			10.83–30,565.43 2868.14 ± 5118.29			
		BCF_B_	0.0010–0.6444 0.0499 ± 0.1402	0.0001–80.43 13.66 ± 22.09	0.0010–1.2981 0.1020 ± 0.2611	0.0002–1.1101 0.0949 ± 0.2294
			0.0001–80.43 3.476 ± 12.51			
	Mouth to the reservoir	P	7.12–698.65 352.84 ± 345.72	32.48–654.36 343.37 ± 310.86	20.09–28.99 24.48 ± 4.17	26.01–875.95 451.01 ± 424.64
			7.12–875.95 292.93 ± 353.44			
		BCF_W_	32.83–3772.12 1891.48 ± 1858.31	271.56–3346.98 1804.33 ± 1525.24	68.34–89.27 78.68 ± 9.81	316.79–8925.21 4592.84 ± 4273.36
			32.83–8925.21 2091.83 ± 2935.64			
		BCF_B_	0.0056–0.0110 0.0083 ± 0.0027	0.2034–35.69 17.6765 ± 17.4764	0.0013–0.0387 0.0197 ± 0.0184	0.0129–0.0517 0.0324 ± 0.0193
			0.0013–35.69 4.4342 ± 11.6107			
	The whole	P	5.73–2159.75 384.25 ± 522.58	5.96–1825.44 392.19 ± 488.35	2.11–62.78 14.91 ± 12.94	4.74–2812.89 489.60 ± 654.14
			2.11–2812.89 320.24 ± 517.41			
		BCF_W_	22.78–17,473.74 2831.38 ± 4221.96	68.24–13,333.54 2560.56 ± 3370.69	10.83–358.97 64.26 ± 68.62	62.74–30,565.43 5415.81 ± 7113.98
			10.83–30,565.43 1975.94 ± 6625.33			
		BCF_B_	0.0010–0.6444 0.0508 ± 0.13	0.0001–71.98 13.82 ± 21.27	0.0010–1.2981 0.0978 ± 0.2438	0.0002–1.1102 0.0941 ± 0.2171
			0.0001–80.43 39.03 ± 42.09			
Strzegomka	Below the springs	P	8.13–45.89 26.99 ± 18.64	10.01–521.63 265.84 ± 255.57	12.24–42.99 27.62 ± 15.06	7.74–675.65 341.59 ± 333.82
			7.74–675.65 165.51 ± 253.28			
		BCF_W_	46.09–346.90 195.22 ± 147.68	83.53–4958.48 2510.25 ± 2421.31	65.68–301.49 183.75 ± 116.35	67.65–6734.25 3396.86 ± 3328.15
			46.09–6734.25 1571.52 ± 2500.40			
		BCF_B_	0.0038–0.0258 0.0148 ± 0.0109	0.0004–0.0282 0.0143 ± 0.0139	0.0045–0.0131 0.0088 ± 0.0042	0.0001–0.0100 0.0051 ± 0.0049
			0.0001–0.0282 0.0107 ± 0.0102			
	Tributaries	P	4.23–59.41 22.35 ± 18.68	10.06–2536.87 631.25 ± 896.22	5.59–56.79 24.16 ± 16.30	3.55–3044.54 753.60 ± 1076.80
			3.55–3044.54 357.84 ± 777.59			
		BCF_W_	25.07–415.16 138.72 ± 129.82	78.017–16,534.75 4295.60 ± 5753.44	35.97–385.58 141.07 ± 114.68	26.60–14,347.52 3845.02 ± 5066.66
			25.07–16,534.75 2105.10 ± 4311.41			
		BCF_B_	0.0020–0.0217 0.0091 ± 0.0068	0.0005–0.1553 0.0367 ± 0.0550	0.0019–0.0232 0.0092 ± 0.0070	0.0001–0.0403 0.0104 ± 0.0144
			0.0001–0.1553 0.0163 ± 0.0311			
	Mouth to the reservoir	P	19.44–33.56 26.50 ± 6.91	20.10–503.98 261.98 ± 241.47	25.11–29.54 27.38 ± 1.91	15.10–610.93 313.11 ± 297.48
			15.10–610.93 157.24 ± 232.42			
		BCF_W_	109.98–217.37 163.81 ± 52.14	126.63–4124.19 2112.67 ± 1980.60	142.80–180.8 161.59 ± 15.88	77.28–3372.21 1686.11 ± 1605.54
			77.28–4124.19 1031.04 ± 1550.05			
		BCF_B_	0.0067–0.0092 0.0079 ± 0.0012	0.0010–0.0102 0.0056 ± 0.0050	0.0088–0.0125 0.0107 ± 0.0017	0.0003–0.0099 0.0051 ± 0.0050
			0.0003–0.0125 0.0073 ± 0.0041			
	The whole	P	4.23–59.41 24.19 ± 17.12	10.01–2536.87 484.31 ± 734.19	5.59–56.79 25.50 ± 14.43	3.55–3044.54 583.10 ± 882.82
			3.55–3044.54 279.25 ± 629.04			
		BCF_W_	25.07–415.16 155.04 ± 124.57	78.02–16534.75 3501.94 ± 4772.74	35.969–385.58 153.71 ± 104.58	26.60–14347.52 3323.61 ± 4339.81
			25.07–16,534.75 1783.57 ± 3614.98			
		BCF_B_	0.0020–0.0258 0.010 ± 0.01	0.0004–0.1553 0.0259 ± 0.045	0.0019–0.0232 0.0094 ± 0.0058	0.0001–0.0402 0.0083 ± 0.0119
			0.0001–0.1553 0.0134 ± 0.02			
Bystrzyca	Below the springs	P	12.02–86.53 49.27 ± 37.25	4.52–456.65 230.41 ± 225.88	11.06–96.98 53.78 ± 42.71	2.36–698.45 350.32 ± 347.95
			2.36–698.45 170.94 ± 244.72			
		BCF_W_	64.89–529.23 297.01 ± 232.10	25.64–2759.24 1368.09 ± 1342.24	55.70–499.18 277.35 ± 221.03	17.29–7124.64 35.67 ± 3550.23
			17.29–7124.64 1377.54 ± 2328.33			
		BCF_B_	0.0032–0.0386 0.0209 ± 0.0178	0.0029–0.0829 0.0429 ± 0.0400	0.0037–0.0083 0.0063 ± 0.0021	0.0007–0.1759 0.0882 ± 0.0876
			0.0007–0.1759 0.0396 ± 0.0580			
	Tributaries	P	10.25–45.53 26.73 ± 11.54	2.56–1005.99 232.37 ± 348.95	3.01–43.79 20.75 ± 13.36	1.25–2022.55 565.10 ± 644.67
			1.25–2022.55 233.99 ± 431.20			
		BCF_W_	82.95–518.78 211.68 ± 111.36	15.25–8974.08 2571.14 ± 2884.50	21.76–294.87 139.58 ± 90.54	10.91–16,268.30 4961.98 ± 5542.95
			10.91–16,268.30 1971.10 ± 3702.05			
		BCF_B_	0.0013–0.0278 0.0082 ± 0.0060	0.0002–0.7058 0.1489 ± 0.1907	0.0002–0.0289 0.0075 ± 0.0086	0.0004–0.8087 0.2460 ± 0.2892
			0.0002–0.8087 0.1026 ± 0.2005			
	Mouth to the reservoir	P	35.41–60.15 47.78 ± 12.36	8.63–202.33 105.45 ± 96.82	33.56–55.45 44.47 ± 10.91	5.54–251.67 128.49 ± 122.94
			5.54–251.67 81.55 ± 86.67			
		BCF_W_	411.12–495.81 453.15 ± 41.74	62.39–1642.21 851.14 ± 787.53	231.16–300.73 265.92 ± 30.75	52.60–2279.64 1136.56 ± 1084.59
			52.60–2279.64 676.70 ± 751.64			
		BCF_B_	0.0039–0.0586 0.0313 ± 0.0274	0.0032–0.0671 0.0351 ± 0.0319	0.0030–0.0139 0.0085 ± 0.0054	0.0019–0.1251 0.0635 ± 0.0615
			0.0019–0.1252 0.0346 ± 0.0422			
	The whole	P	10.25–86.53 31.09 ± 18.35	2.56–1005.99 292.28 ± 328.78	3.01–96.98 26.43 ± 21.68	1.25–2022.55 499.96 ± 604.60
			1.25–2022.55 212.44 ± 397.17			
		BCF_W_	64.89–529.23 244.36 ±144.82	15.25–8974.08 2278.84 ± 2693.27	21.76–499.18 165.98 ± 119.73	10.91–16268.30 4440.01 ± 5229.03
			10.91–16,268.30 1782.30 ± 3424.89			
		BCF_B_	0.0013–0.0586 0.0117 ± 0.0138	0.0002–0.7058 0.1266 ± 0.1769	0.0002–0.0289 0.0075 ± 0.0079	0.0004–0.8087 0.2119 ± 0.2696
			0.0002–0.8087 0.0895 ± 0.1826			

P—plant; BCF_W_—aluminum bioaccumulation factor in relation to water; BCF_B_—aluminum bioaccumulation factor in relation to bottom sediments.

**Table 3 ijerph-19-02930-t003:** Aluminum content in aquatic plants (mgAl·kg^−1^ on dry weight basis) and aluminum accumulation in spring and autumn.

	Site	Nysa Szalona		Site	Strzegomka		Site	Bystrzyca	
		Spring	Autumn		Spring	Autumn		Spring	Autumn
P	1	3.15–13.27 8.30 ± 3.79	24.66–562.78 336.26 ± 197.46	1	7.74–12.89 9.74 ± 1.89	42.45–675.65 321.28 ± 282.43	1	2.36–12.02 7.49 ± 4.14	86.52–698.45 334.39 ± 257.54
BCF_W_		14.63–107.77 76.94 ± 37.24	141.03–6045.50 3008.06 ± 2181.28		46.09–93.40 68.17 ± 14.85	298.55–6734.25 3074.86 ± 2825.53		17.29–64.92 41.09 ± 19.97	497.07–7124.64 2713.99 ± 2696.23
BCF_B_		0.0002–0.2224 0.0540 ± 0.09	0.4134–25.9640 6.3215 ± 10.64		0.0001–0.0050 0.0022 ± 0.01	0.0100–0.0282 0.0192 ± 0.0078		0.0007–0.0055 0.0027 ± 0.0014	0.0082–0.1759 0.0764 ± 0.0632
P	2	5.96–13.98 10.11 ± 2.85	15.07–456.72 289.67 ± 165.34	2	3.55–13.86 7.99 ± 4.09	5.59–589.65 266.75 ± 260.41	2	2.40–15.33 8.31 ± 5.59	43.09–852.63 372.50 ± 345.16
BCF_W_		45.56–123.96 74.59 ± 29.31	99.15–5367.66 2590.09 ± 1860.43		25.07–82.31 51.57 ± 23.97	35.97–5245.72 2313.77 ± 2283.04		17.56–107.58 52.73 ± 37.22	227.28–8685.02 3181.98 ± 3425.10
BCF_B_		0.0001–0.0426 0.0114 ± 0.02	0.0265–14.2383 3.6325 ± 6.07		0.0001–0.0046 0.0018 ± 0.01	0.0019–0.0235 0.0093 ± 0.01		0.0002–0.0058 0.0029 ± 0.01	0.0031–0.3284 0.1458 ± 0.14
P	3	4.89–12.48 8.62 ± 3.33	2.88–2812.89 1700.17 ± 1042.23	3	5.94–15.70 11.04 ± 4.43	27.57–3044.54 1410.21 ± 1391.81	3	1.25–42.64 21.99 ± 20.09	25.32–756.55 336.62 ± 317.63
BCF_W_		22.70–131.07 76.38 ± 50.32	19.39–30,565.43 15196.28 ± 10,796.41		26.60–94.89 57.91 ± 26.53	143.35–16,534.75 7726.89 ± 7604.01		10.91–289.33 148.53 ± 135.29	144.63–6295.96 2643.00 ± 2556.23
BCF_B_		0.0002–0.0135 0.0039 ± 0.01	0.0010–71.9811 18.43 ± 30.91		0.0001–0.0051 0.0023 ± 0.01	0.0108–0.1553 0.0569 ± 0.06		0.0002–0.0268 0.0086 ± 0.01	0.0023–0.2572 0.1048 ± 0.11
P	4	2.38–15.46 9.89 ± 5.03	62.26–907.59 651.79 ± 342.28	4	15.10–25.86 20.30 ± 3.52	29.00–610.93 294.18 ± 265.56	4	4.55–10.26 6.93 ± 2.10	6.95–1499.99 558.79 ± 613.82
BCF_W_		11.39–182.74 92.09 ± 69.78	354.21–9984.44 5735.40 ± 3471.46		49.31–141.27 94.90 ± 34.04	378.44–5303.83 2385.56 ± 2078.02		31.26–83.07 50.39 ± 19.78	42.65–13,636.27 4855.80 ± 5458.77
BCF_B_		0.0002–0.0087 0.0029 ± 0.01	0.0304–44.27 11.24 ± 19.06		0.0003–0.0098 0.0047 ± 0.01	0.0144–0.0392 0.0229 ± 0.01		0.0002–0.0050 0.0018 ± 0.01	0.0044–0.6044 0.2420 ± 0.25
P	5	9.52–24.89 15.40 ± 5.85	19.46–1146.79 776.85 ± 443.04	5	15.10–25.89 20.30 ± 3.52	29.00–610.93 294.181 ± 265.56	5	3.25–16.53 9.88 ± 6.09	25.34–701.65 281.30 ± 278.73
BCF_W_		32.63–185.02 118.72 ± 54.61	108.34–13,985.24 6954.68 ± 4870.60		77.28–150.24 117.74 ± 25.57	174.35–4124.19 1944.34 ± 1771.01		25.26–131.06 74.07 ± 47.85	163.81–6558.71 2467.29 ± 2669.07
BCF_B_		0.0024–1.1128 0.3085 ± 0.46	0.0018–80.43 34.62 ± 35.46		0.0003–0.0092 0.0048 ± 0.0040	0.0066–0.0125 0.0098 ± 0.0021		0.0004–0.0060 0.0030 ± 0.01	0.0011–0.2761 0.1022 ± 0.11
P	6	4.45–14.65 7.90 ± 4.06	21.34–652.88 360.19 ± 250.55				6	2.07–13.43 5.96 ± 4.47	38.22–1142.60 469.18 ± 462.21
BCF_W_		20.53–121.91 57.04 ± 38.55	127.22–7574.05 3135.21 ± 2782.74					15.22–93.84 40.85 ± 30.82	288.48–10,929.71 4306.13 ± 4325.73
BCF_B_		0.0002–0.4849 0.1234 ± 0.21	0.0034–4.3189 1.0909 ± 1.86					0.0002–0.0061 0.0026 ± 0.01	0.0013–0.8087 0.2761 ± 0.32
P	7	3.24–20.94 10.99 ± 6.80	6.12–1756.99 1190.62 ± 693.31				7	2.39–32.42 15.68 ± 12.99	5.22–1364.60 562.75 ± 579.67
BCF_W_		13.65–249.57 97.81 ± 92.54	23.59–19,048.06 10,771.61 ± 6871.59					18.93–170.23 95.41 ± 73.17	41.73–12,167.75 4761.89 ± 5023.88
BCF_B_		0.0011–0.5429 0.1896 ± 0.22	0.0033–13.9559 3.5327 ± 6.02					0.0002–0.0148 0.0068 ± 0.01	0.0012–0.6069 0.2343 ± 0.25
P	8	4.84–18.52 13.44 ± 5.1458	16.43–873.98 589.70 ± 337.54				8	3.57–15.64 7.56 ± 4.73	30.74–2022.55 773.12 ± 823.47
BCF_W_		19.93–172.80 121.37 ± 58.79	44.73–9543.77 5016.18 ± 3387.07					31.02–152.21 64.84 ± 50.50	223.81–16,268.30 6326.52 ± 6601.39
BCF_B_		0.0009–0.5095 0.1659 ± 0.20	0.0082–3.9993 1.0164 ± 1.72					0.0005–0.0063 0.0027 ± 0.01	0.0023–0.7707 0.3725 ± 0.37
P	9	2.91–13.78 9.07 ± 4.62	5.54–502.99 343.94 ± 198.14				9	2.33–25.63 11.59 ± 9.56	32.09–681.52 301.67 ± 278.90
BCF_W_		13.26–145.15 72.24 ± 50.54	14.77–6207.73 3028.52 ± 2181.75					22.08–182.32 83.26 ± 66.19	236.87–5394.65 2385.76 ± 2220.92
BCF_B_		0.0016–0.8808 0.2416 ± 0.37	0.0029–2.5686 0.6507 ± 1.11					0.0006–0.0102 0.0053 ± 0.01	0.0058–0.2785 0.1295 ± 0.12
P	10	5.85–14.26 10.66 ± 3.13	8.45–315.98 194.78 ± 115.28				10	5.54–60.15 32.43 ± 25.41	33.56–251.67 130.67 ± 97.74
BCF_W_		31.19–368.31 129.37 ± 127.35	24.51–2777.56 1519.65 ± 1005.70					52.60–495.81 227.08 ± 182.66	231.16–2279.64 1126.31 ± 832.02
BCF_B_		0.0032–0.3712 0.1169 ± 0.15	0.0030–2.5596 0.6426 ± 1.11					0.0019–0.0586 0.0167 ± 0.0242	0.0039–0.1252 0.0525 ± 0.0483
P	11	9.51–32.89 18.44 ± 8.88	21.33–1337.87 873.57 ± 505.78						
BCF_W_		85.75–203.99 127.99 ± 47.12	74.77–14,697.91 7074.12 ± 5110.67						
BCF_B_		0.0031–1.2981 0.3371 ± 0.55	0.0085–65.5874 16.1847 ± 27.95						
P	12	8.49–14.75 12.06 ± 2.23	16.43–542.56 364.39 ± 204.36						
BCF_W_		35.40–135.92 86.27 ± 38.53	48.38–5891.04 2701.08 ± 2098.58						
BCF_B_		0.0033–0.0575 0.0235 ± 0.02	0.0061–14.8612 3.69 ± 6.37						
P	13	2.11–32.60 17.38 ± 13.24	26.45–821.91 556.51 ± 311.61						
BCF_W_		10.83–311.54 152.44 ± 122.77	75.89–9962.59 3906.23 ± 3666.38						
BCF_B_		0.0034–0.0460 0.0179 ± 0.02	0.0103–1.0699 0.2803 ± 0.45						
P	14	7.12–32.56 21.57 ± 9.39	28.44–875.95 564.28 ± 320.14						
BCF_W_		32.83–322.10 175.21 ± 125.64	87.30–8925.21 4008.45 ± 3142.18						
BCF_B_		0.0056–0.2041 0.0651 ± 0.08	0.0013–35.69 8.80 ± 15.21						

P—plant; BCF_W_—aluminum bioaccumulation factor in relation to water; BCF_B_—aluminum bioaccumulation factor in relation to bottom sediments.

**Table 4 ijerph-19-02930-t004:** Metal pollution index (MPI) of aquatic plants with aluminum.

Year	Nysa Szalona	Bystrzyca	Strzegomka
2015	72.17	26.46	18.07
2016	98.61	48.64	101.77
2017	10.21	18.62	21.33
2018	103.35	48.95	85.90
Average	71.08	35.67	56.76

**Table 5 ijerph-19-02930-t005:** Metal pollution index (MPI) of aquatic plants, bottom sediments, and water with aluminum [24,25].

	Nysa Szalona			Bystrzyca			Strzegomka		
Material	Plant	Sediment	Water	Plant	Sediment	Water	Plant	Sediment	Water
MPI	71.08	277.26	0.0903	35.67	189.41	0.0396	56.76	50.46	0.0826
Pollution degree	very high	highest	no effect	high	highest	no effect	very high	high	no effect

**Table 6 ijerph-19-02930-t006:** Metal pollution index (MPI) of aquatic plants with aluminum—mouth, springs.

	Nysa Szalona	Bystrzyca	Strzegomka
Below the springs	39.19	37.35	39.34
Pollution degree	high		
Mouth to the reservoir	79.36	41.98	46.81
Pollution degree	very high	high	

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
