# Peer review of "Aluminum Bioaccumulation in Reed Canary Grass (Phalaris arundinacea L.) from Rivers in Southwestern Poland"

_ijerph, 2022, doi:10.3390/ijerph19052930_

Round 1

Reviewer 1 Report

The study seems relevant and with a good scientific basis. However, the overall manuscript is poorly written if we consider the number of small sentences and how they flow ... this makes the manuscript very hard for readers. Nevertheless, I still consider that has the potential to be published after major revision.

Author Response

We have edited the text so that it would be easier to follow.

Reviewer 2 Report

The authors present a study evaluating accumulation of aluminum by reed canary grass (Phalaris arundinacea L.) in three rivers of southwestern Poland. Reed canary grass was collected from varying numbers of sites in different locations along the three rivers during two seasons between 2015-2018. The aluminum concentrations reported in the reed canary grass were similar to that of previous research that have studied the same or similar rivers in the area, and the authors suggest that consequences of modernization of riverbeds may show delayed effects in relation to pollution.

The authors present an impressive data set related to aluminum accumulation in plants from rivers in Poland and the data collected are consistent with previous studies. Overall, the manuscript is well written and organized, but the significance of the study is not clearly presented. In the introduction, the authors provide a brief paragraph on the toxicological effects of aluminum on humans, but this information does appear to relate to the specific objectives of the study. Similarly, there is very little context for the study provided in the objective statement. Therefore, it is unclear if this study was done to document pollution in relation to human health, plant health, ecosystem health, or the influence of urbanization on aluminum pollution. The authors also provide very little information related to the land use surrounding these rivers that may be influencing the accumulation of aluminum within the plants, such that the final conclusion that modernization may show delayed impacts on aluminum pollution feels unfounded. Without specific context as to why this study was conducted or what these data mean for the area or the scientific field, the manuscript lacks any relevant impact. Therefore, it is recommended that the manuscript be resubmitted after major revision. Specific comments are below.

Specific Comments

Lines 58-64: The specific context for this information being included is not clear. The data collected in the current study do not appear to be related to any documented human health concerns in the area, such that it is not clear what relevance human health has in the context of this study. Similarly, this topic is not ever mentioned again in the manuscript, such that this information seems irrelevant.

Lines 88-90: The specific objective of the study lacks context. Was aluminum accumulation documented to relate this information to human health, pollution due to land use changes, plant health, or ecosystem health? Without this context, the data do not have any impact for the reader.

Lines 99-101: The figure caption should include labels A, B, and C, not I, II, and III to match the figure.

Line 279-281: This sentence references data that were collected in 2011 but the methods describe data being collected between 2015-2018. Were these data collected from a previous study or is the date incorrectly represented?

Table 4: “In Total” appears that it should be “Average”, as the values represented are an average.   

Author Response

The authors present a study evaluating accumulation of aluminum by reed canary grass (Phalaris arundinacea L.) in three rivers of southwestern Poland. Reed canary grass was collected from varying numbers of sites in different locations along the three rivers during two seasons between 2015-2018. The aluminum concentrations reported in the reed canary grass were similar to that of previous research that have studied the same or similar rivers in the area, and the authors suggest that consequences of modernization of riverbeds may show delayed effects in relation to pollution. The authors present an impressive data set related to aluminum accumulation in plants from rivers in Poland and the data collected are consistent with previous studies. Overall, the manuscript is well written and organized, but the significance of the study is not clearly presented. In the introduction, the authors provide a brief paragraph on the toxicological effects of aluminum on humans, but this information does appear to relate to the specific objectives of the study. – part of the text concerning toxicological effects of aluminum on humans has been removed

Similarly, there is very little context for the study provided in the objective statement. Therefore, it is unclear if this study was done to document pollution in relation to human health, plant health, ecosystem health, or the influence of urbanization on aluminum pollution. the objective of the study has been rewritten

The authors also provide very little information related to the land use surrounding these rivers that may be influencing the accumulation of aluminum within the plants, such that the final conclusion that modernization may show delayed impacts on aluminum pollution feels unfounded. Without specific context as to why this study was conducted or what these data mean for the area or the scientific field, the manuscript lacks any relevant impact. – the following has been included in the 2.1. Study area section: “The land use along all three rivers is quite similar: rural areas, wastelands, forests, small towns with currently limited economic management, poorly regulated water and sewage management, limited to larger towns. A significant difference between the rivers is the landform (lowland, upland, and mountain catchment), which can affect leaching of aluminum from soils.”

Therefore, it is recommended that the manuscript be resubmitted after major revision. Specific comments are below.

Specific Comments

Lines 58-64: The specific context for this information being included is not clear. The data collected in the current study do not appear to be related to any documented human health concerns in the area, such that it is not clear what relevance human health has in the context of this study. Similarly, this topic is not ever mentioned again in the manuscript, such that this information seems irrelevant. – this paragraph has been removed

Lines 88-90: The specific objective of the study lacks context. Was aluminum accumulation documented to relate this information to human health, pollution due to land use changes, plant health, or ecosystem health? Without this context, the data do not have any impact for the reader. – the objective of the study has been expanded

Lines 99-101: The figure caption should include labels A, B, and C, not I, II, and III to match the figure. – it has been changed

Line 279-281: This sentence references data that were collected in 2011 but the methods describe data being collected between 2015-2018. Were these data collected from a previous study or is the date incorrectly represented? – the date should be 2018, it has been corrected in the text

Table 4: “In Total” appears that it should be “Average”, as the values represented are an average.  – it has been changed

Reviewer 3 Report

REVIEW REPORT

Title: OK

Abstract: It is expected that the abstract should have numerical and statistical data for better understanding. Authors are required to rewrite the abstract (in accordance to the author’s instruction).

Keywords: OK

Introduction:

Line No. 46 “Aluminum is commonly found in plants and its content ranges from X0 to X00 mg·kg- 1.” Justify and rewrite it.

Line No.65-85: Rewrite it with valid references.

Line No. 86: Which rivers?

Materials and Methods:

Line No. 160-167- this description is not necessary rather the economic importance of the target plant is necessary. Authors are required to do the needful.

What is the sampling method? Composite/ random….explain.

Use separate headings for For the determination of aluminum content. Line 180

Statistical analysis of the results: OK

Result and Discussion:

Table 2. Aluminum content (mgAl·kg-1) (on fresh or dry weight basis) disclose it

Same for table.3 and n=?

Why columns were left blank?

Why there is a difference in Al content among surface and benthic zones?

What will be possible environmental consequences of Al accumulation in reed canary grass? Explain it from Local-Regional-Global prospective.

In my opinion, the authors have conducted elaborate investigation. However, the MS can’t be accepted in present form. There are some issues need to be addressed and incorporated into the MS. I am suggesting a MAJOR REVISION.

Author Response

Title: OK

Abstract: It is expected that the abstract should have numerical and statistical data for better understanding. Authors are required to rewrite the abstract (in accordance to the author’s instruction). we have shortened the abstract according to the author’s instruction, however, due to space limits and data abundance, we have decided not to include numerical and statistical data as this is not obligatory according to the  instruction

Keywords: OK

Introduction:

Line No. 46 “Aluminum is commonly found in plants and its content ranges from X0 to X00 mg·kg- 1.” Justify and rewrite it. – values and reference have been included

Line No.65-85: Rewrite it with valid references. – references have been introduced

Line No. 86: Which rivers? – not specified rivers, “the” has been removed

Materials and Methods:

Line No. 160-167- this description is not necessary rather the economic importance of the target plant is necessary. Authors are required to do the needful. – it has been changed according to the suggestion, description has been removed and the following has been introduced: “Monocotyledonous emergent plants (helophytes), reed canary grass (Phalaris arundinacea L.), growing on river banks were collected for the study [15]. The plant used for the study is a fast growing, perennial plant characterized by an early season growth, high physiological tolerance and large opportunities of applications. It has excellent properties from combustion point of view, and thus is valued due to its possible application as energy crop. It can be also used in the production of biogas, ethanol, paper or pulp as well as for manufacturing of chemical raw materials [16-18].”

What is the sampling method? Composite/ random….explain. – the samples were collected randomly, this information has been included in the text

Use separate headings for For the determination of aluminum content. Line 180 – new heading has been used

Statistical analysis of the results: OK

Result and Discussion:

Table 2. Aluminum content (mgAl·kg-1) (on fresh or dry weight basis) disclose it – aluminum content was determined on dry weight basis, which has been added in the table title

Same for table.3 and n=? - aluminum content was determined on dry weight basis, which has been added in the table title, n = 232 which is pointed out in Methods part

Why columns were left blank? - There are 3 rivers there and each has a different number of sites, hence the longest one - Nysa Szalona - occupies all the columns and the others correspondingly less.

Why there is a difference in Al content among surface and benthic zones? - Larger amounts of Al in the bottom zone indicate the origin of aluminum from bottom sediments. This is probably the case here. There could be even more Al if the environment was more acidic and there was more possibility for it to migrate from the sediment into the water.

What will be possible environmental consequences of Al accumulation in reed canary grass? Explain it from Local-Regional-Global prospective. – The following paragraph has been added to conclusions section: “Aluminum accumulation in reed canary grass involves the retention of aluminum in plant tissues. This is particularly important in areas where water is collected for drinking. Aluminum ions present in water as a primary product that is later used as a raw material in water production plants should not reach high concentrations. Therefore, it is important that the aquatic vegetation present in water bodies - here rivers - both submerged and emerged, such as the reed canary grass discussed here, can act as a bioaccumulator and absorb aluminum compounds. The amounts that are thus accumulated in nature will not need to be removed by humans during the water cycle. In addition, the accumulation of aluminum in rivers from their sources to the mouth reduces the flow of aluminum in the catchment. In this case, it inhibits the flow of aluminum in the Oder River basin and, further, in the Baltic Sea basin. Hence, the role of hydromacrophytes is so important in the absorption of pollutants, including aluminum under discussion here.”

Round 2

Reviewer 2 Report

The authors have provided context for the study, which has greatly improved the clarity and relevance of the manuscript. The manuscript should be accepted for publication.

Reviewer 3 Report

Editor will tell you.